

# The rate of information transfer as a measure of ocean-atmosphere interactions

David Docquier[1], Stéphane Vannitsem[1], and Alessio Bellucci[2]

[1]Royal Meteorological Institute of Belgium, Brussels, Belgium
[2]Italian National Research Council, Institute of Atmospheric Science and Climate, Bologna, Italy

**Correspondence:** D. Docquier (david.docquier@meteo.be)

**Abstract.** Exchanges of energy between the ocean and atmosphere are of large importance in regulating the climate system. Here we apply for the first time a relatively novel approach, the rate of information transfer, to quantify interactions between the ocean surface and lower atmosphere over the period 1988-2017 at monthly time scale. More specifically, we investigate dynamical dependencies between sea-surface temperature (SST), SST tendency and turbulent heat flux in satellite observations.

We find a strong two-way influence between SST / SST tendency and turbulent heat flux in many regions of the world, with largest values in eastern tropical Pacific and Atlantic oceans, as well as in western boundary currents. The total number of regions with a significant influence of turbulent heat flux on SST and on SST tendency is reduced when considering the three variables, suggesting an overall stronger ocean influence compared to the atmosphere. We also find a relatively strong influence of turbulent heat flux taken one month before on SST. Additionally, an increase in the magnitude of the rate of information

transfer and in the number of regions with significant influence is observed when looking at interannual and decadal time scales, compared to monthly time scale.

## 1   Introduction

The climate on Earth is strongly affected by energy exchanges between the ocean and atmosphere. The ocean absorbs a large amount of solar energy and releases part of this energy to the atmosphere. In turn, the atmosphere modifies the ocean state

through changes in winds, humidity and temperature. The classical view is that the slowly-changing upper ocean is modulated by the high-frequency atmospheric variability (Hasselmann, 1976; Frankignoul and Hasselmann, 1977). While this paradigm has been successful in explaining the variability in sea-surface temperature (SST) and surface heat flux over large parts of the ocean, it has been challenged over ocean regions characterized by intense mesoscale activity, such as western boundary currents and Antarctic Circumpolar Current (Chelton et al., 2004; Brachet et al., 2012; Kirtman et al., 2012; Bishop et al.,

2017; Roberts et al., 2017; Small et al., 2020; Bellucci et al., 2021).

Chelton et al. (2004) used 25 km resolution satellite radar scatterometer measurements over 1999-2003 and revealed the existence of persistent small-scale features in wind stress. According to Chelton et al. (2004), much of that mesoscale variability is attributable to SST modification. In particular, they found that surface wind speed is locally higher over warm water and lower over cool water, i.e. a positive correlation that is opposite to the one found at large scale. Bishop et al. (2017) found that





monthly-scale lead-lag correlations between SST, SST tendency and turbulent heat flux (THF) allow to discriminate between
atmospheric-driven variability and ocean-led variability, using results from a stochastic energy balance model (Wu et al.,
2006) designed to represent ocean-atmosphere interactions, as well as satellite observations over 1985-2013. In their analysis,
when the SST variability is dominated by atmospheric weather, SST tendency is negatively correlated with THF anomalies,
while when the SST fluctuations are driven by intrinsic ocean processes (ocean weather), the correlation between SST and

THF is positive. Following this approach, they showed that eddy-rich regions associated with pronounced SST gradients,
such as western boundary currents, are characterized by ocean-driven SST variability, while less dynamically active, open
ocean regions, characterized by weaker SST gradients, exhibit an atmosphere-driven SST variability. These two regimes are
reproduced by both eddy-parameterized (∼1° spatial resolution) and eddy-permitting (∼0.25°) coupled global climate models
(Bellucci et al., 2021), with an increased ocean resolution leading to a substantially improved representation of SST and THF

cross-covariance patterns.

According to a commonly accepted interpretation, the positive SST-THF zero-lag correlation (identifying an ocean-driven
regime) indicates a damping role of the turbulent heat fluxes on the SST anomalies generated by ocean dynamics, while the
negative SST tendency-THF correlation (identifying an atmosphere-driven regime) is attributed to the ocean surface cooling
determined by the release of heat from the ocean to the atmosphere. However, the presence of such correlations does not firmly

demonstrate causal influences between these variables, as correlation does not mean causation. Thus, the use of a dedicated
causal method is of crucial importance to corroborate these findings. Several causal inference frameworks have been developed
in the past years to identify such causal links (Granger, 1969; Liang and Kleeman, 2005; Sugihara et al., 2012; Krakovská et al.,
2018; Palûs et al., 2018; Runge et al., 2019).

The Liang-Kleeman information flow method allows to identify the direction and magnitude of the cause-effect relationships

between variables (Liang and Kleeman, 2005). It is based on the rate of information transfer in dynamical systems and is
rigorously derived from the propagation of information entropy between variables (Liang, 2016). It has initially been developed
for two-variable systems (Liang, 2014) and has recently been extended to multivariate systems (Liang, 2021). Compared to
other causal inference frameworks, the rate of information transfer is a relatively simple index to compute one-way and/or two-
way dependencies between variables. This novel method has been successfully applied to several climate studies, e.g. causal

influences between greenhouse gases and global mean surface temperature (Stips et al., 2016; Jiang et al., 2019), dynamical
dependencies between a set of observables and the Antarctic surface mass balance (Vannitsem et al., 2019), soil moisture - air
temperature interactions in China (Hagan et al., 2019), prediction of El Niño Modoki (Liang et al., 2021), causal links between
climate indices in the North Pacific and Atlantic regions (Vannitsem and Liang, 2022), and identification of potential drivers
of Arctic sea-ice changes (Docquier et al., 2022).

In our study, we analyze upper ocean - lower atmosphere interactions using the rate of information transfer developed
by Liang (2021). More specifically, we check the two-way influences between SST / SST tendency and THF at the air-sea
interface in satellite observations. Thus, our study allows to go one step further than previous studies (Bishop et al., 2017;
Bellucci et al., 2021), which have mainly focused on lead-lag correlation analyses, by identifying causal links between these



variables. Section 2 presents the data and methods used in this analysis. Section 3 provides the main results of our study and

places them in the overall context. Our conclusions are presented in Sect. 4.

## 2   Data and Methods

### 2.1   Data

We use version 3 of the Japanese Ocean Flux Data Sets with the Use of Remote-Sensing Observations (J-OFURO3; Tomita et al., 2019; Tomita, 2020). This dataset uses multiple satellite data to estimate surface fluxes between the ocean and atmosphere

over sea ice-free regions with a resolution of $0.25°$. It makes use of passive microwave radiometers and scatterometers available from 1988 to 2017 (see Tomita et al. (2019) for further details). From this dataset, we extract monthly mean latent and sensible heat fluxes, as well as SST. The latter is computed as an ensemble median obtained from various global SST products.

We also use the SeaFlux Data Products to estimate the observational uncertainty. It consists of estimates of ocean surface latent and sensible heat fluxes, among other variables (Roberts et al., 2020). It relies on the use of Special Sensor Microwave

Imager (SSM/I) and Special Sensor Microwave Imager Sounder (SSMIS) over the period 1988-2018 (we use 1988-2017 to be consistent with J-OFURO3). The SST is also available for this dataset and is computed using the Reynolds Optimally-Interpolated Version 2.0 (Reynolds et al., 2007). As for J-OFURO3, SeaFlux data are available on a $0.25°$ grid and we extract monthly mean latent and sensible heat fluxes and SST.

In the main text, we only show results from J-OFURO3, as results obtained with SeaFlux are largely consistent. The latter

are presented in Appendix B.

### 2.2   Methods

Our analysis involves three variables, namely SST, SST tendency and THF, following the approach of Bishop et al. (2017) and Bellucci et al. (2021). The choice of these three specific variables is based on the stochastic energy balance model of Wu et al. (2006). As explained in Section 1, lead-lag covariances between these three variables are used as a way to diagnose

ocean-driven and atmosphere-driven regimes. The goal of our analysis is to go beyond the correlation / covariance relationships identified by Bishop et al. (2017) and Bellucci et al. (2021) and check the causal links between SST, SST tendency and THF. THF is defined as the sum of latent heat flux and sensible heat flux and is expressed in W m$^{-2}$ (positive upwards), as in Bishop et al. (2017) and Bellucci et al. (2021). SST tendency is computed via a central difference approximation of SST (expressed in °C) using a time step of one month (following Bishop et al., 2017) and is expressed in °C month$^{-1}$.

We compute the rate of information transfer between SST, SST tendency and THF using monthly data from 1988 to 2017. The absolute rate of information transfer from variable $X_j$ to variable $X_i$ is computed following Liang (2021):

$$T_{j \to i} = \frac{1}{\det \mathbf{C}} \cdot \sum_{k=1}^{d} \Delta_{jk} C_{k,di} \cdot \frac{C_{ij}}{C_{ii}}, \tag{1}$$



where $\mathbf{C}$ is the covariance matrix, $d$ is the number of variables, $\Delta_{jk}$ are the cofactors of $\mathbf{C}$, $C_{k,di}$ is the sample covariance between $X_k$ and the Euler forward difference approximation of $dX_i/dt$ ($dt$ is the time step and equals one month in our study), $C_{ij}$ is the sample covariance between $X_i$ and $X_j$, $C_{ii}$ is the sample variance of $X_i$.

To assess the relative importance of the different cause-effect relationships, we compute the relative rate of information transfer from variable $X_j$ to variable $X_i$ following Liang (2021):

$$\tau_{j \to i} = \frac{T_{j \to i}}{Z_i}, \tag{2}$$

where $Z_i$ is the normalizer, computed as follows:

$$Z_i = \sum_{k=1}^{d} |T_{k \to i}| + \left| \frac{dH_i^{noise}}{dt} \right|, \tag{3}$$

where the first term on the right-hand side represents the information flowing from all the $X_k$ to $X_i$ (including the influence of $X_i$ on itself), and the last term is the effect of noise, computed following Liang (2021).

When $\tau_{j \to i}$ is statistically different from 0 (either positive or negative), $X_j$ has an influence on $X_i$, while if $\tau_{j \to i} = 0$ there is no influence. A value of $|\tau| = 100\%$ indicates that $X_j$ has the maximum influence on $X_i$. A positive (negative) value of $\tau_{j \to i}$ means that the variability in $X_j$ makes the variability in $X_i$ more uncertain (certain), i.e. it increases (decreases) the variability in $X_i$ (Appendix A; Fig. A1). Statistical significance of $\tau_{j \to i}$ is computed via bootstrap resampling with replacement of all terms included in equations (1)-(3) using 500 realizations. These boostrap realizations are combined together using the False Discovery Rate (FDR) from Wilks (2016) with a significance level of 5% to account for the multiplicity of tests.

## 3 Results and discussion

Bishop et al. (2017) identified a strong positive zero-lag covariance between SST and THF at monthly time scale (over 1985-2013) in western boundary currents, Agulhas Return Current and eastern tropical Pacific, using the OAFlux dataset (1° resolution) for THF and the NOAA OISST dataset (0.25° resolution) for SST (see their Fig. 3b). They also found a strong negative zero-lag covariance between SST tendency and THF over many regions of the globe, with largest values at mid-latitudes (see their Fig. 3e). Using the J-OFURO3 dataset (0.25° resolution), we find similar results in terms of zero-lag covariance (Fig. B1). Additionally, when mapping the Pearson correlation coefficient instead of the covariance, we find a strong positive correlation between SST and THF in many regions of the world, with largest values in eastern tropical Pacific and Atlantic regions, and in western boundary currents (Fig. 1(a)). A strong negative correlation between SST tendency and THF is also identified in most parts of the world, with the exception of a relatively narrow band along the equator and in western boundary currents (Fig. 1(b)).

Bishop et al. (2017) and Bellucci et al. (2021) showed that regions of high SST gradient and THF (such as the Gulf Stream) are characterized by an ocean-driven regime. In these regions, the SST-THF zero-lag covariance is positive, suggesting that ocean processes drive SST anomalies, and the SST tendency-THF lead-lag covariance is anti-symmetric (positive covariance at lag -1 and negative covariance at lag +1). On the contrary, an anti-symmetric SST-THF lead-lag covariance and a negative SST





**Figure 1.** Pearson correlation coefficient (a) between sea-surface temperature (SST) and turbulent heat flux (THF) and (b) between SST tendency (SSTt) and THF, based on J-OFURO3 satellite observations. Black contours are drawn around regions with a statistically significant correlation coefficient (FDR 5%; Student's t-test).





tendency-THF zero-lag covariance are typical of an atmosphere-driven regime (such as in the North Atlantic subtropical gyre).
In the latter case, the release of heat flux from the ocean to the atmosphere acts to cool the upper ocean. While this approach is interesting to identify whether the SST variability is driven by ocean or atmosphere processes, it does not precisely indicate whether the SST causally influences THF or the other way round. In our study, we quantify the causal relationships between SST / SST tendency and THF using the rate of information transfer from Liang (2021).

### 3.1 The two-dimensional (2D) case

If we only take into account SST and THF (i.e. two-dimensional system, hereafter referred to as 2D) in the computation of the rate of information transfer, we find that many regions are characterized by relatively strong two-way influences between SST and THF (Fig. 2). The spatial distribution of the rate of information transfer is relatively similar to the one of correlation coefficient between SST and THF (Fig. 1(a)), with largest values (either positive or negative) in eastern tropical Pacific and Atlantic regions, western boundary currents and many parts of the Southern Hemisphere. Interestingly, the rate of information
transfer is mainly positive for the influence of SST on THF (Fig. 2(a)), while negative values dominate for the reverse influence (Fig. 2(b)). This suggests that SST variability generally increases THF variability, while THF variability mainly constrains SST variability. Also, the regions where the influence from SST on THF is strongest are also characterized by a strong influence from THF on SST (in absolute value).

Regarding the SST tendency-THF relationship, using only these two variables in the computation of the rate of information
transfer also provides relatively strong two-way influences (either positive or negative) in many regions of the world (Fig. 3). The spatial distribution is more contrasted than with the SST-THF relationship, as both positive and negative values are now present in both directions, especially for the influence of SST tendency on THF (Fig. 3a). The information transfer from SST tendency to THF is characterized by positive southwest-northeast bands in the North Atlantic and North Pacific, positive northwest-southeast bands in the South Atlantic and South Pacific, and negative values between these regions (Fig. 3(a)). The
information transfer from THF to SST tendency shows a relatively symmetrical behavior to that from SST tendency to THF, with positive (negative) values where the reverse information transfer is negative (positive) (Fig. 3(b)). This indicates that in regions where the variability in SST tendency increases (decreases) the variability in THF, the variability in THF decreases (increases) the variability in SST tendency.

In summary, the 2D analysis shows additional information compared to previous lead-lag correlation studies (Bishop et al.,
2017; Bellucci et al., 2021). In particular, we find that the ocean surface influences the lower atmosphere not only in strong boundary currents, but also in many other regions of the world (Fig. 2(a) and 3(a)). In turn, the lower atmosphere (via surface heat fluxes) influences the ocean surface not only at mid-latitudes, but also in tropical regions and western boundary currents (Fig. 2(b) and Fig. 3(b)). Our results are in line with Bach et al. (2019), who also find significant two-way influences between the upper ocean and lower atmosphere in many regions of the world using the Granger causality, but with some methodological
differences (daily time scale and other atmospheric fields). This shows that the lead-lag covariance analysis, while interesting in identifying a particular ocean-driven or atmospheric-led regime, is not sufficient to accurately quantify causal links between



**(a)** $\tau_{SST \rightarrow THF}$ 1988-2017 - J-OFURO3

**(b)** $\tau_{THF \rightarrow SST}$ 1988-2017 - J-OFURO3

**Figure 2.** Relative rate of information transfer $\tau$ (a) from sea-surface temperature (SST) to turbulent heat flux (THF) and (b) from THF to SST, based on J-OFURO3 satellite observations, when two variables are considered. Black contours are drawn around regions with a statistically significant transfer of information (FDR 5%; 500 bootstrap samples).



**Figure 3.** Relative rate of information transfer $\tau$ (a) from sea-surface temperature tendency (SSTt) to turbulent heat flux (THF) and (b) from THF to SSTt, based on J-OFURO3 satellite observations, when two variables are considered. Black contours are drawn around regions with a statistically significant transfer of information (FDR 5%; 500 bootstrap samples).





the upper ocean and lower atmosphere. Importantly, our analysis (like the ones from Bishop et al. (2017) and Bellucci et al. (2021)) applies to the monthly time scale, and we will show results beyond this specific time scale in Sect. 3.4.

## 3.2 The three-dimensional (3D) case

The analysis done so far with two variables may provide a false impression of a two-way influence emerging due to the absence of a set of hidden variables. Thus, we computed the rate of information transfer based on the three variables analyzed in Bishop et al. (2017) and Bellucci et al. (2021), namely SST, SST tendency and THF (hereafter referred to as 3D). The 3D case provides additional sources of information compared to the 2D case and should thus be preferred in terms of result interpretations.

In the 3D case, the influence from SST to THF (Fig. 4(a)) is very similar to the 2D case (Fig. 2(a)). This is logical since
SST has no significant influence on SST tendency (Fig. B2), so all the information from SST goes to THF, demonstrating the robustness of the approach. However, a much reduced number of regions shows a significant rate of information transfer from THF to SST (Fig. 4(b)) compared to the 2D case (Fig. 2(b)). This reduction is due to the fact that we now take SST tendency into account in the computation of the rate of information transfer: part of the information transfer from THF also goes into SST tendency, as we will see below. Despite this reduction in the number of regions with significant transfer of information,
the eastern tropical Pacific and Atlantic regions still show a strong negative rate of information transfer, suggesting that THF variability constrains SST variability in these regions (Fig. 4(b)). Interestingly, some regions (such as the Agulhas Return Current) show a positive rate of information transfer from THF to SST in the 3D case (Fig. 4(b)), while it is negative or close to 0 in the 2D case (Fig. 2(b)).

In the 3D case, the rate of information transfer from SST tendency to THF (Fig. 5(a)) is very similar to the 2D case (Fig. 3(a))
for the same reason as for the SST-THF influence. However, a much reduced number of regions shows a significant transfer of information from THF to SST tendency (Fig. 5(b)) compared to the 2D case (Fig. 3(b)). Similarly as for the influence of THF on SST, this is due to the inclusion of a third variable: part of the information transfer from THF also goes into SST. Nevertheless, we still find regions of significant influence, e.g. negative values in the North Atlantic and northeastern Pacific, and positive values in tropical regions.

Thus, computing the rate of information transfer based on the three variables of interest (SST, SST tendency and THF) somehow partitions the total influence of the lower atmosphere (THF) into a contribution to SST (Fig. 4(b)) and another contribution to SST tendency (Fig. 5(b)). Additionally, the total number of regions with a significant influence from THF to SST and from THF to SST tendency (both combined) clearly decreases compared to the 2D case. As the influences from SST to THF and from SST tendency to THF remain strong in the 3D case (almost unchanged compared to the 2D case), this goes
in favor of a stronger ocean influence compared to the atmosphere at monthly time scale, especially in extra-tropical regions.

In western boundary currents, we find large values of the rate of information transfer from SST to THF (Fig. 4(a)), suggesting a strong ocean influence, in agreement with Bishop et al. (2017). However, in extra-tropical regions far away from western boundary currents, such as in the central North Atlantic, we also find a strong transfer of information from SST (Fig. 4(a)) and from SST tendency (Fig. 5(a)) to THF, generally stronger than the reverse influence (from THF to SST and to SST tendency).
This puts somewhat in question previous findings that suggest an atmospheric-driven SST variability in these regions (Bishop



**Figure 4.** Relative rate of information transfer $\tau$ (a) from sea-surface temperature (SST) to turbulent heat flux (THF) and (b) from THF to SST, based on J-OFURO3 satellite observations, when three variables are considered (SST, SST tendency and THF). Black contours are drawn around regions with a statistically significant transfer of information (FDR 5%; 500 bootstrap samples).





**Figure 5.** Relative rate of information transfer $\tau$ (a) from sea-surface temperature tendency (SSTt) to turbulent heat flux (THF) and (b) from THF to SSTt, based on J-OFURO3 satellite observations, when three variables are considered (SST, SST tendency and THF). Black contours are drawn around regions with a statistically significant transfer of information (FDR 5%; 500 bootstrap samples).





et al., 2017; Bellucci et al., 2021) and this shows that lead-lag covariance analyses should be supplemented by causality studies. The use of the SeaFlux observational dataset provides results in broad agreement with J-OFURO3 (Figs. B3-B4), which confirms the robustness of our findings.

The extension to additional variables can of course be performed to refine the analysis further. We can do this by including other fields, higher-order tendencies, or lagged fields. A first step toward such type of analysis is provided below. In our study, we prefer to keep the three fields used in the stochastic energy balance model from Wu et al. (2006) and analyzed by Bishop et al. (2017) and Bellucci et al. (2021), i.e. SST, SST tendency and THF. We will therefore focus on the use of a lagged field (Sect. 3.3) as well as the analysis of interannual to decadal variability (Sect. 3.4).

### 3.3    Lagged transfer of information

Due to the inertia of the ocean mixed layer, the SST does not necessarily respond directly to changes in THF (Deser et al., 2003; Shi et al., 2022). To take this effect into account, we added a fourth variable to our analysis: THF leading SST by one month, hereafter referred to as 'THF(-1)'. The rate of information transfer has been applied to lagged variables in a previous study to predict El Niño Modoki based on solar activity (Liang et al., 2021).

We find that there is a significant positive rate of information transfer from THF(-1) to SST in eastern tropical Pacific and Atlantic regions, western boundary currents (Gulf Stream and Kuroshio Extension) and Agulhas Return Current, as well as negative values in other parts of the world (Fig. 6a). Thus, the lagged analysis shows that THF taken one month before strongly controls SST variability, especially in northern extra-tropical regions where this influence is almost absent in the original 3D case (Fig. 4b). There is also additional information provided by this fourth variable in the transfer of information from THF(-1) to SST tendency (Fig. 6b), but this is mostly restricted to eastern tropical Pacific and Atlantic (negative values). The influence from SST / SST tendency to THF(-1) is found to be very close to 0 and almost everywhere not significant (Fig. B5), which confirms the robustness of the method, as causality cannot go back in time.

Interestingly, most regions showing a positive rate of information transfer from THF(-1) to SST (Fig. 6a) also have a positive lead-lag covariance between THF(-1) and SST (Fig. 3c of Bishop et al. (2017)). Bishop et al. (2017) show that these regions have a symmetric lead-lag structure between SST and THF, which is characteristic of an ocean-driven regime. However, these similarities between the rate of information transfer and the lead-lag covariance disappear when we look at the relationships between THF(-1) and SST tendency. According to Bishop et al. (2017), the strongest values of lead-lag covariance between THF(-1) and SST tendency appear in western boundary currents (negative values; Fig. 3f of Bishop et al. (2017)), while the rate of information transfer is significant only in eastern tropical Pacific and Atlantic (negative values; Fig. 6b).

### 3.4    Interannual to decadal variability

All previous results are based on monthly mean outputs. Thus, our results are valid at monthly time scale. In order to figure out what happens at interannual and decadal time scales, we take the 12-month and 120-month running mean SST and THF, respectively, and re-compute the rate of information transfer in the 3D case (SST, SST tendency and THF). This approach





**(a)** $\tau_{THF(-1)\rightarrow SST}$ 1988-2017 - J-OFURO3

**(b)** $\tau_{THF(-1)\rightarrow SSTt}$ 1988-2017 - J-OFURO3

**Figure 6.** Relative rate of information transfer $\tau$ (a) from turbulent heat flux at lag -1 (THF(-1); THF leading SST by 1 month) to sea-surface temperature (SST) and (b) from THF(-1) to SST tendency (SSTt), based on J-OFURO3 satellite observations, when four variables are considered (SST, SST tendency, THF and THF(-1)). Black contours are drawn around regions with a statistically significant transfer of information (FDR 5%; 500 bootstrap samples).





is similar to the one used in Vannitsem and Liang (2022), who found differences in the rate of information transfer between climate indices depending on the time scale used.

At interannual time scale (12-month running mean), the influence of SST on THF encompasses approximately the same regions as at monthly time scale but with a general increase in the magnitude of the rate of information transfer (Fig. 7a). We also find regions that have a negative rate of information transfer, such as in the western North Atlantic Ocean, whereas such regions are not present with monthly means. The reverse influence of THF on SST provides a more contrasted pattern compared to the original 3D case, with a reduced number of regions with negative values along the equator but the additional
presence of regions with positive values (Fig. 7b).

At decadal time scale (120-month running mean), almost the whole globe is covered by significant information transfer between SST and THF in the two directions (Fig. 8). Also, the magnitude of the rate of information transfer clearly increases at this time scale compared to interannual and decadal time scales. These results suggest that ocean-atmosphere interactions become more pronounced at larger time scale.

## 230  4   Conclusions

In summary, we find that the rate of information transfer provides a more detailed quantification of dependencies between SST, SST tendency and turbulent heat flux (THF) than previous classical correlation-covariance studies. We do not argue that causal methods should replace covariance analyses, but they should rather be used as a complement in order to get a better understanding of physical interactions between variables. We show that the ocean surface (SST and SST tendency) strongly
drives changes in the lower atmosphere (THF) and that the lower atmosphere also has an important influence on the ocean surface in many regions of the world. This result is somewhat different from what has been found in covariance analyses, in which ocean-driven regimes exist in western boundary currents and atmospheric-led regimes dominate in the open ocean (Bishop et al., 2017; Bellucci et al., 2021). It is however supported by another recent analysis, using the Granger causality, which shows that many regions of the world present a significant two-way influence between the lower atmosphere and the
upper ocean, with a stronger ocean influence in tropical regions and a stronger atmospheric impact in the extra-tropics (Bach et al., 2019). As the latter study presents methodological differences, i.e. it focuses on the daily time scale and uses different atmospheric variables, we need to be cautious in comparing it to our analysis. In any case, our interpretation of these results is that ocean-atmosphere interactions are more complex than presented by classical covariance analyses.

Furthermore, we find that the influence of THF is partitioned between SST and SST tendency if we consider the three
variables together (3D case), so that the single impact of either THF on SST or THF on SST tendency is decreased in the 3D case compared to the 2D case. Also, the number of regions with a significant rate of information transfer from THF to SST and from THF to SST tendency (combined) is smaller than the one from SST to THF and from SST tendency to THF (combined) in the 3D case. This suggests an overall stronger upper ocean influence compared to the lower atmosphere. However, when adding THF taken one month before to take the lagged effect into account, we find that this variable has a relatively strong





**Figure 7.** Relative rate of information transfer $\tau$ (a) from sea-surface temperature (SST) to turbulent heat flux (THF) and (b) from THF to SST, based on J-OFURO3 satellite observations, when three variables are considered (SST, SST tendency and THF) and using 12-month running mean (interannual variability). Black contours are drawn around regions with a statistically significant transfer of information (FDR 5%; 500 bootstrap samples).



**Figure 8.** Relative rate of information transfer $\tau$ (a) from sea-surface temperature (SST) to turbulent heat flux (THF) and (b) from THF to SST, based on J-OFURO3 satellite observations, when three variables are considered (SST, SST tendency and THF) and using 120-month running mean (decadal variability). Black contours are drawn around regions with a statistically significant transfer of information (FDR 5%; 500 bootstrap samples).





influence on SST in a large portion of the globe. Finally, a larger time scale (going from monthly to interannual and decadal) provides larger values of the rates of information transfer between the three variables.

In our study, we only considered the ocean surface, but several studies have shown that variations in the ocean heat content are controlled by both air-sea fluxes and ocean heat transport convergence, with a more important role for the latter with a deeper integration of ocean heat and with higher-resolution climate models (Roberts et al., 2017; Small et al., 2020). Also,

only observations were considered here. Thus, extending our analysis to the ocean heat budget terms in climate models would provide further insights into the causal influences between the ocean and atmosphere. This is of large importance as ocean-atmosphere interactions constitute an important regulator of our climate. Finally, from a theoretical perspective, additional investigations of the role of hidden and lagged variables should be performed.

*Code and data availability.* J-OFURO3 observational data (Tomita et al., 2019; Tomita, 2020) are available on https://j-ofuro.isee.nagoya-u.

ac.jp/en/dataset/entry-323.html. SeaFlux observational data (Roberts et al., 2020) are accessible from NASA (https://cmr.earthdata.nasa.gov/search/concepts/C1995869798-GHRC_DAAC.html). The Python scripts to produce the figures of this article are available on Zenodo: https://zenodo.org/record/7074861 (Docquier, 2022).

## Appendix A:  Impact of the rate of information transfer on the variability

As explained in the main text (Sect. 2.2), a positive value of the relative rate of information transfer $\tau_{j \to i}$ means that the

variability in $X_j$ increases the variability in $X_i$, while a negative value means that the variability in $X_j$ decreases the variability in $X_i$. More broadly, an increase in the rate of information transfer from $X_j$ to $X_i$ leads to an increases in the variability of $X_i$. We demonstrate this by computing the rate of information transfer $\tau_{j \to i}$ from variable $X_j$ to variable $X_i$ (based on equation (3) in the main text), using a three-dimensional stochastic linear system of equations:

$$dX_1 \;=\; (a_{11}\,X_1 + a_{12}\,X_2 + a_{13}\,X_3)\,dt + 0.1\,dW_1$$

$$dX_2 \;=\; (a_{21}\,X_1 + a_{22}\,X_2 + a_{23}\,X_3)\,dt + 0.1\,dW_2$$

$$dX_3 \;=\; (a_{31}\,X_1 + a_{32}\,X_2 + a_{33}\,X_3)\,dt + 0.1\,dW_3, \tag{A1}$$

where $X_1$, $X_2$ and $X_3$ are the three variables, $a_{kl}$ are the different coefficients, $t$ is time and varies between 0 and 100 with 100,000 time steps ($\Delta t = 0.001$), and $W_1$, $W_2$ and $W_3$ represent normal random noises (standard Wiener process). We set $a_{11}$ = $a_{22} = a_{33}$ = -1, and we vary the six other coefficients one by one with 5 different values between -1 and 1 (-1, -0.5, 0, 0.5,

1). When varying one of the six coefficients, we set the other five coefficients to a fixed value ($a_{12} = a_{13} = 0.5$ and $a_{21} = a_{23} = a_{31} = a_{32} = 0$).

We solve the linear system (A1) using the Euler-Maruyama method, and 40 different values of the random noise are taken in order to take into account the uncertainty related to the rate of information transfer. The variance in each variable is compared to the rate of information transfer from any other variable to this variable to test our hypothesis. Results show that when the

rate of information transfer from $X_j$ to $X_i$ increases, the variance in $X_i$ increases (Fig. A1).



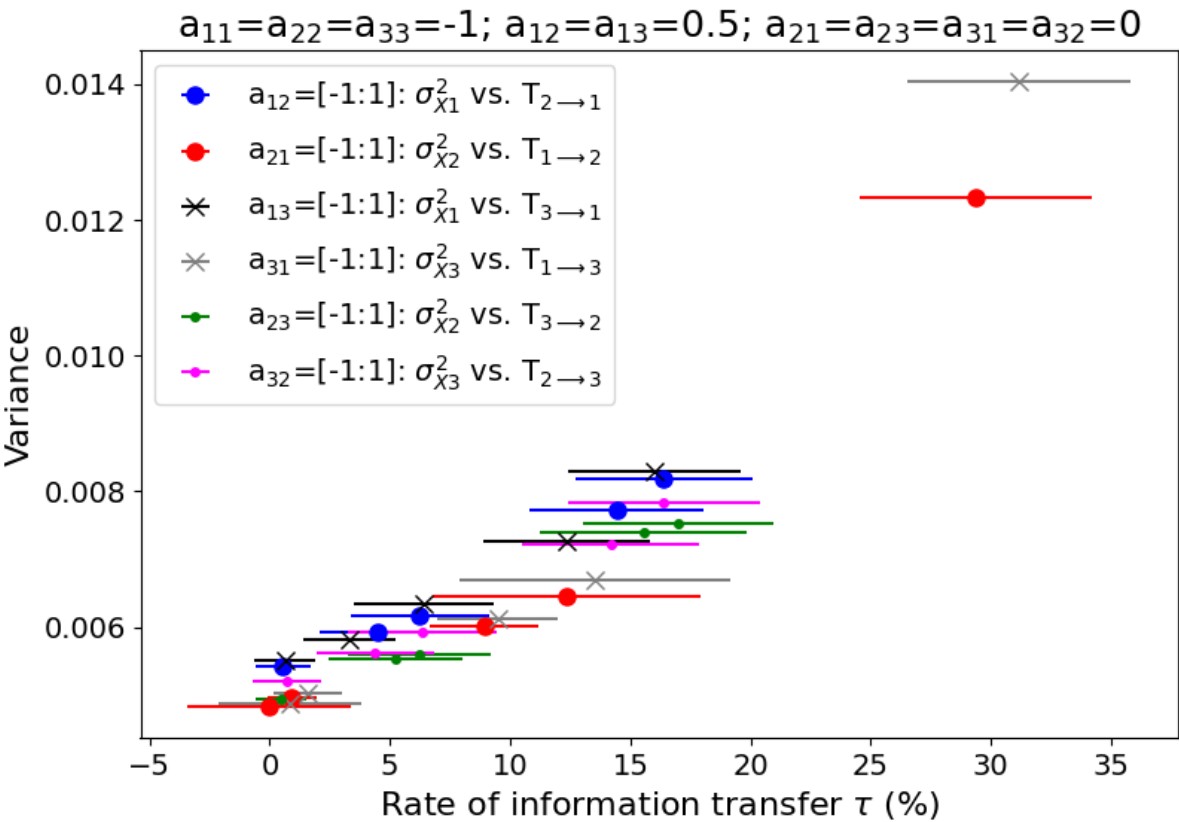

**Figure A1.** Variance $\sigma^2$ of a variable as a function of the relative rate of information transfer $\tau$ from any other variable to this variable using the linear system of equations (A1). The error bars show the 95% confidence intervals of the rates of information transfer.



## Appendix B: Supplementary figures

**Figure B1.** Covariance (a) between sea-surface temperature (SST) and turbulent heat flux (THF) and (b) between SST tendency (SSTt) and THF, based on J-OFURO3 satellite observations.



**(a)** $\tau_{SST \rightarrow SSTt}$ 1988-2017 - J-OFURO3

**(b)** $\tau_{SSTt \rightarrow SST}$ 1988-2017 - J-OFURO3

**Figure B2.** Relative rate of information transfer $\tau$ (a) from sea-surface temperature (SST) to SST tendency and (b) from SST tendency to SST, based on J-OFURO3 satellite observations, when three variables are considered (SST, SST tendency and THF). None of the grid points shows a statistically significant transfer of information (FDR 5%; 500 bootstrap samples).



**(a)**     $\tau_{SST \to THF}$ 1988-2017 - SeaFlux

**(b)**     $\tau_{THF \to SST}$ 1988-2017 - SeaFlux

**Figure B3.** Relative rate of information transfer $\tau$ (a) from sea-surface temperature (SST) to turbulent heat flux (THF) and (b) from THF to SST, based on SeaFlux satellite observations, when three variables are considered (SST, SST tendency and THF). Black contours are drawn around regions with a statistically significant transfer of information (FDR 5%; 500 bootstrap samples).



**(a)**



**(b)**

**Figure B4.** Relative rate of information transfer $\tau$ (a) from sea-surface temperature tendency (SSTt) to turbulent heat flux (THF) and (b) from THF to SSTt, based on SeaFlux satellite observations, when three variables are considered (SST, SST tendency and THF). Black contours are drawn around regions with a statistically significant transfer of information (FDR 5%; 500 bootstrap samples).




**(a)** $\tau_{SST \rightarrow THF(-1)}$ 1988-2017 - J-OFURO3

**(b)** $\tau_{SSTt \rightarrow THF(-1)}$ 1988-2017 - J-OFURO3

**Figure B5.** Relative rate of information transfer $\tau$ (a) from sea-surface temperature (SST) to turbulent heat flux one month before (THF(-1)) and (b) from SST tendency to THF(-1), based on J-OFURO3 satellite observations, when four variables are considered (SST, SST tendency, THF and THF(-1)). Black contours are drawn around regions with a statistically significant transfer of information (FDR 5%; 500 bootstrap samples).



*Author contributions.* DD wrote the manuscript with contributions from all co-authors. DD, SV and AB designed the science plan. DD analyzed satellite data and produced the figures. All authors participated in the interpretation of results and provided useful comments to help improve the analysis.

*Competing interests.* The authors declare that they have no conflict of interest.

*Acknowledgements.* We thank X. S. Liang for useful discussions related to the rate of information transfer. We thank C. Frankignoul for the very helpful feedback provided following one of our presentations. DD, SV and AB are supported by ROADMAP (Role of ocean dynamics and Ocean-Atmosphere interactions in Driving cliMAte variations and future Projections of impact-relevant extreme events; https://jpi-climate.eu/project/roadmap/), a coordinated JPI-Climate/JPI-Oceans project. DD and SV received funding from the Belgian Federal
Science Policy Office under contract B2/20E/P1/ROADMAP. AB received funding from MUR - Italian Ministry of University and Research (D.D. n.1316, 08/06/2021).



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
