# Peer review of "The rate of information transfer as a measure of ocean-atmosphere interactions"

_EGUsphere, 2022_

## Referee Comment (RC2)

Review of

Title: The rate of information transfer as a measure of ocean-atmosphere interactions

Author(s): David Docquier et al.

MS No.: egusphere-2022-942

MS type: Research article

The authors apply the Liang (or Liang-Kleeman) information flow (LIF thereafter) in order to quantify interactions between the ocean surface and lower atmosphere over the period 1988–2017 at monthly time scale. They investigate dynamical dependencies between sea-surface temperature (SST), SST tendency and turbulent heat flux in satellite observations and find a **strong two-way influence** between SST / SST tendency and turbulent heat flux in many regions of the world.

LIF is a very interesting approach, independent of all other causality methods known in the literature. It was analytically derived for dynamical systems and its general application requires the knowledge of the underlying equations of the studied dynamical systems. The only form, available for experimental data without the knowledge of the equations, was derived **for linear systems**. In spite of this, the approach has recently been applied for data from apparently nonlinear systems, relying on a few numerical examples by the original author in which LIF was applied to nonlinear systems.

In general, there are cases when linear approaches can extract correct causal relations from nonlinear data. One example is the causality in highly nonGaussian data from space weather area [1]. In the study [1], however, the analysis started by a general nonlinear approach – conditional mutual information (CMI thereafter, a.k.a. transfer entropy). In the next step, the response to time reversal suggested [according to study 2] that although the data were nonlinear, the observed causality is a sort of linear information transfer. The latter was confirmed by the application of the linearized version of CMI as well as by LIF, both giving consistent results with the nonlinear CMI. Applying a linear approach, LIF in this MS, alone, can be dangerous.

I will explain my concern using the well-known causality benchmark of unidirectionally coupled Rössler systems, described, e.g., in [3]. Figure 1 in this review, left panel, illustrates the successful application of LIF to the unidirectionally coupled Rössler systems, presented originally by X. San Liang. It took me some time to reproduce this result. The key to obtain the correct distinction of the direction of coupling is using very high sampling frequency, which is about 6000 samples per period (or pseudoperiod in this chaotic system), see Fig. 1 right panel. With this oversampling, the nonlinear dynamics is locally linearized. In Fig. 1. left panel, we can see that LIF in the direction of coupling ("causal direction") increases with the increase of the coupling strength ε, while the LIF in the non-causal direction (the direction with no coupling, i.e. from the effect to the cause) remains on the zero value. A slightly disturbing fact is that the ε-dependence ignores the transient to synchronization (cf Fig. 4 in [3]), however, LIF in this case correctly identifies the causal direction.

The result is different when LIF is applied to the unidirectionally coupled Rössler systems sampled with "usual" frequency, with about 20 samples per period, which is sufficient for inferring causality using nonlinear methods [3]. We can see in Fig. 2 that LIF in the causal direction nonmonotonically increases with the increase of the coupling strength ε, however, the LIF in the non-causal direction does the same, just with the negative sign. That is, when the sampling does not allow linearization of the problem, LIF detects information flow also in the direction where there is no connection, just its value is negative. The plot of LIF as function of coupling strength ε results in a symmetric figure (Fig. 2, left panel) with the zero axis as the axis of symmetry, meaning that LIF(x->y)=−LIF(y->x).

[Figure]

Fig. 1.: Left panel: LIF applied to coupled Rössler systems with very high sampling frequency, illustrated in the right panel.

[Figure]

Fig. 2.: Left panel: LIF applied to coupled Rössler systems with usual sampling frequency, illustrated in the right panel.

The results presented in the current manuscript, e.g. MS Fig. 2 for the relation between SST and THF reflect the same symmetric pattern: LIF(SST->THF) is positive, marked by red color in the used color scale, while LIF(THF->SST) is negative, marked by blue color in the used color scale, and the red and blue patterns in parts (a) and (b) are the same. This is the same results as in the case of the unidirectionally coupled Rössler systems with usual sampling frequency, presented in our Fig. 2 above, i.e. the authors obtained that LIF(SST->THF)=−LIF(THF->SST). For any further discussion of the results presented in this MS, the authors should provide an evidence, based on an independent, nonlinear method, that the "symmetric information flow" (interpreted as a **strong two-way influence)** between SST or tSST and THF is indeed a physical phenomenon and not just a failure of the linear LIF applied to nonlinear data, as observed in the case of the unidirectionally coupled Rössler systems above.

[1] Manshour, Pouya, et al. "Causality and information transfer between the solar wind and the magnetosphere–ionosphere system." Entropy 23.4 (2021): 390.

[2] Paluš, Milan, et al. "Causality, dynamical systems and the arrow of time." Chaos: An Interdisciplinary Journal of Nonlinear Science 28.7 (2018): 075307.

[3] Paluš, M., & Vejmelka, M. (2007). Directionality of coupling from bivariate time series: How to avoid false causalities and missed connections. Physical Review E, 75(5), 056211.

M. Paluš, December 10, 2022

---

## Author Comment (AC1)

**Reply to Anonymous Referee #1**

**Manuscript egusphere-2022-942**
**"The rate of information transfer as a measure of ocean-atmosphere interactions"**
**D. Docquier, S. Vannitsem, A. Bellucci**

**We would like to thank the reviewer for his/her helpful comments. Below we present** the reviewer's comments (in black) **and our point-by-point replies (including changes in the paper) (in bold blue). Line numbers correspond to the revised version.**

**Comments**

In physical oceanography it is believed that wind stress drives the ocean, while in dynamical meteorology the ocean surface is treated as a bottom boundary that influences the atmosphere. The interaction between the sea surface temperature (SST) and wind stress, respectively characterizing the sea and the atmosphere at the interface, has become of enormous interest. In this paper, the authors applied a causality analysis which is built on a firm physical ground, in contrast to other statistical formalisms, to the study of this problem, and obtained intriguing new results. Specifically, they found that that the ocean surface (SST and SST tendency) strongly drives changes in the lower atmosphere (THF) and that the lower atmosphere also has an important influence on the ocean surface in many regions of the world, different from the traditional view that ocean-driven regimes largely exist in western boundary currents and atmospheric-led regimes dominate in the open ocean. In recognition of the importance of the finding, I hence recommend publication of this manuscript. The following are just some points that the authors may pay some attention.

**We thank the reviewer for the very positive comment. We have taken all comments from the reviewer into account and have revised our paper accordingly.**

l.1 & l.13, True. But in this paper, the usage of information transfer/information flow (IF) in studying the interaction is actually more fundamental. It is the exchange of entropy/information rather than energy. In statistical physics, entropy plays a role in distributing energy.

**In the beginning of the abstract and introduction, we refer to the general physical relationship between the ocean and atmosphere, rather than the statistical relationship (based on entropy). In order not to have any confusion between information transfer and "physical" energy exchanges, we changed the terminology "exchanges of energy" by "exchanges of mass, momentum and energy" (L1 and L14).**

l.97, the last term may also represent the effect from unobserved processes.

**This is correct, we have now clarified this in the paper (L101).**

l.100, While mathematically this is correct in terms of Shannon entropy, you may want to be more cautious in interpreting the sign, as it actually may not be explained using the well-known physics.

**We agree with the reviewer that we need to be cautious with this interpretation, but this is the interpretation provided by Liang (2014), so we prefer to keep this terminology in the paper. As stated in Liang (2014), a positive (negative) value of $T_{j \to i}$ means that the variability in $X_j$**

**makes the variability in $X_i$ more uncertain (certain); we have added reference to Liang (2014) in this sentence (L104). We also make reference to Appendix A and Figure A1 in the paper (L105), where we show that when $T_{j \to i}$ increases, the variance in $X_i$ also increases.**

ll. 130-135. ). "This suggests that SST variability generally increases THF variability, while THF variability mainly constrains SST variability." This is good. But be cautious.

**We thank the reviewer for the word of caution. Please see our response to the previous comment.**

ll.189-190. To include more additional variables, make sure they are not nearly parallel; otherwise the singularity of the covariance matrix could numerically deteriorate the result.

**We thank the reviewer for this comment and have added it to the paper (L194-195).**

Section 3.3. In studying lagged transfer of information, be careful that only the IF in one way makes sense—Causality cannot be from the future to the past.

**We agree with the reviewer and that is exactly why we checked the other direction to verify the robustness of the method. We found that the rate of information transfer from SST / SST tendency to THF(-1) is statistically not significant (Fig. B5). This information is provided at L210-212.**

ll.226-229. Indeed ocean-atmosphere interactions become more pronounced at larger time scale. So these results do make sense.

**We thank the reviewer for this comment.**

---

## Author Comment (AC2)

**Reply to Referee #2 (Milan Paluš)**

**Manuscript egusphere-2022-942**
**"The rate of information transfer as a measure of ocean-atmosphere interactions"**
**D. Docquier, S. Vannitsem, A. Bellucci**

**We would like to thank the reviewer M. Paluš for his helpful comments. Below we present** the reviewers' comments (in black) **and our point-by-point replies (including changes in the paper) (in bold blue).**

**Comments**

The authors apply the Liang (or Liang-Kleeman) information flow (LIF thereafter) in order to quantify interactions between the ocean surface and lower atmosphere over the period 1988-2017 at monthly time scale. They investigate dynamical dependencies between sea-surface temperature (SST), SST tendency and turbulent heat flux in satellite observations and find a **strong two-way influence** between SST / SST tendency and turbulent heat flux in many regions of the world.

LIF is a very interesting approach, independent of all other causality methods known in the literature. It was analytically derived for dynamical systems and its general application requires the knowledge of the underlying equations of the studied dynamical systems. The only form, available for experimental data without the knowledge of the equations, was derived **for linear systems**. In spite of this, the approach has recently been applied for data from apparently nonlinear systems, relying on a few numerical examples by the original author in which LIF was applied to nonlinear systems.

In general, there are cases when linear approaches can extract correct causal relations from nonlinear data. One example is the causality in highly nonGaussian data from space weather area [1]. In the study [1], however, the analysis started by a general nonlinear approach – conditional mutual information (CMI thereafter, a.k.a. transfer entropy). In the next step, the response to time reversal suggested [according to study 2] that although the data were nonlinear, the observed causality is a sort of linear information transfer. The latter was confirmed by the application of the linearized version of CMI as well as by LIF, both giving consistent results with the nonlinear CMI. Applying a linear approach, LIF in this MS, alone, can be dangerous.

I will explain my concern using the well-known causality benchmark of unidirectionally coupled Rössler systems, described, e.g., in [3]. Figure 1 in this review, left panel, illustrates the successful application of LIF to the unidirectionally coupled Rössler systems, presented originally by X. San Liang. It took me some time to reproduce this result. The key to obtain the correct distinction of the direction of coupling is using very high sampling frequency, which is about 6000 samples per period (or pseudoperiod in this chaotic system), see Fig. 1 right panel. With this oversampling, the nonlinear dynamics is locally linearized. In Fig. 1. left panel, we can see that LIF in the direction of coupling ("causal direction") increases with the increase of the coupling strength ε, while the LIF in the non-causal direction (the direction with no coupling, i.e. from the effect to the cause) remains on the zero value. A slightly disturbing fact is that the ε-dependence ignores the transient to synchronization (cf Fig. 4 in [3]), however, LIF in this case correctly identifies the causal direction.

The result is different when LIF is applied to the unidirectionally coupled Rössler systems sampled with "usual" frequency, with about 20 samples per period, which is sufficient for inferring causality using nonlinear methods [3]. We can see in Fig. 2 that LIF in the causal direction nonmonotonically increases with the increase of the coupling strength ε, however, the LIF in the non-causal direction does the same, just with the negative sign. That is, when the sampling does not allow linearization of the problem, LIF detects information flow also in the direction where there is no connection, just its value is negative. The plot of LIF as function of coupling strength ε results in a symmetric figure (Fig. 2, left panel) with the zero axis as the axis of symmetry, meaning that LIF(x→y)=−LIF(y→x).

Fig. 1.: Left panel: LIF applied to coupled Rössler systems with very high sampling frequency, illustrated in the right panel. → **See comment from M. Paluš** for displaying this figure.

Fig. 2.: Left panel: LIF applied to coupled Rössler systems with usual sampling frequency, illustrated in the right panel. → **See comment from M. Paluš** for displaying this figure.

The results presented in the current manuscript, e.g. MS Fig. 2 for the relation between SST and THF reflect the same symmetric pattern: LIF(SST->THF) is positive, marked by red color in the used color scale, while LIF(THF->SST) is negative, marked by blue color in the used color scale, and the red and blue patterns in parts (a) and (b) are the same. This is the same results as in the case of the unidirectionally coupled Rössler systems with usual sampling frequency, presented in our Fig. 2 above, i.e. the authors obtained that LIF(SST->THF)=−LIF(THF->SST). For any further discussion of the results presented in this MS, the authors should provide an evidence, based on an independent, nonlinear method, that the "symmetric information flow" (interpreted as a strong two-way influence) between SST or tSST and THF is indeed a physical phenomenon and not just a failure of the linear LIF applied to nonlinear data, as observed in the case of the unidirectionally coupled Rössler systems above.

[1] Manshour, Pouya, et al. "Causality and information transfer between the solar wind and the magnetosphere–ionosphere system." Entropy 23.4 (2021): 390.

[2] Paluš, Milan, et al. "Causality, dynamical systems and the arrow of time." Chaos: An Interdisciplinary Journal of Nonlinear Science 28.7 (2018): 075307.

[3] Paluš, M., & Vejmelka, M. (2007). Directionality of coupling from bivariate time series: How to avoid false causalities and missed connections. Physical Review E, 75(5), 056211.

M. Paluš, December 10, 2022

**We thank the reviewer M. Paluš for his highly detailed comment, which helped us to improve our manuscript. We have revised the paper by adding a new sub-section (Section 3.5), which highlights the limitation of the linear approach. We also provide our reply to the reviewer's comment below.**

**We agree with the reviewer that the method of Liang (2014) has been designed for linear systems and caution needs to be taken in using this approach for nonlinear systems, such as the climate system. However, as the reviewer mentions, the approach has been successfully validated for some nonlinear synthetic problems (e.g. Liang, 2014; Liang, 2018; Liang, 2021). It has also been applied to several real-case climate studies, in which results have a physical**

sense (e.g. Docquier et al., 2022). We now provide a detailed explanation of this aspect in Section 3.5, inspired by the reviewer's comment.

We have applied the conditional mutual information (CMI) using the algorithm from Mesner & Shalizi (2021), based on nearest neighbors, on the SST - SSTt - THF data used in our study (monthly data from 1988 to 2017). We have done so for two different cases: a case mimicking the 2D (or 2-variable) case of our study and a case resembling the 3D (or 3-variable) case of our study. We have verified our results with another method based on Gaussian copula using the algorithm from Ince et al. (2016) and find very similar results.

In the 2D case, we have computed:
- $CMI_{SST \rightarrow THF}$, which is the causal link from SST at time *t* to THF one time lag further *t+1* given THF at time *t*, noted *I{SST(t) ; THF(t+1) | THF(t)}* in the CMI framework (Paluš et al., 2018), see Fig. Aa below;
- $CMI_{THF \rightarrow SST}$, which is the causal link from THF at time *t* to SST one time lag further *t+1* given SST at time *t*, noted *I{THF(t) ; SST(t+1) | SST(t)}* in the CMI framework, see Fig. Ab below. Figure A below shows that the causal influence from THF to SST is much stronger than from SST to THF and that, while the former shows strong spatial variations (large values in western boundary currents, tropical regions and North Pacific), the latter is much more homogeneous. Please note that this result is not fully comparable to the 2D case using the Liang index in our study (Fig. 2), as we include time-delayed information in the CMI computation.

However, we show in our study that it is important to take all three variables into account, i.e. SST, SST tendency (SSTt) and THF, when computing causal influences (Section 3.2). Thus, we have also computed the CMI for the 3D case:
- $CMI_{SST \rightarrow THF | SSTt}$, which is the causal link from SST to THF given SST tendency, noted *I(SST ; THF | SSTt}* in the CMI framework, see Fig. Ba below;
- $CMI_{THF \rightarrow SST | SSTt}$, which is the causal link from THF to SST given SST tendency, noted *I(THF ; SST | SSTt}* in the CMI framework, see Fig. Bb below.
Figure B below shows that causal influences are identical in both directions, which can be verified by eq. (7) of the CMI formula in Paluš et al. (2018) if we switch *X* and *Y*. Also, the largest influences appear in eastern tropical Pacific and Atlantic regions and western boundary currents (Fig. B), in agreement with the Liang index (Fig. 4a in the paper). Results of the CMI method thus appear somewhat closer to the Liang index in the 3D case; however, the fact that the two influences are identical with the CMI represents a severe limitation of the method when considering three variables.

Additionally, we have computed the Liang index (in both bivariate and multivariate cases) for the unidirectionally coupled Rössler systems using the same parameters as Paluš & Vejmelka (2007) and Paluš et al. (2018) and a coupling strength varying between 0 and 0.25 with an increment of 0.025. We have used different numbers of samples per pseudo-period to make a similar test as the reviewer. In the bivariate case (only considering *x1* and *y1*), we obtain a very similar result to Fig. 2 of the reviewer, i.e. a symmetrical influence in both directions, whatever the number of samples per pseudo-period, see Figs. C-D below (where we show results for 20 and 6000 samples). In the multivariate case (considering all 6 variables of the two Rössler systems), results depend on the number of samples per pseudo-period, see Figs. E-I below (where we show results for 20, 60, 80, 200 and 6000 samples). However, the almost perfect symmetry we had in the bivariate case disappears in the multivariate case. For 20 and 40 samples per pseudo-period, $T_{y1 \rightarrow x1} > T_{x1 \rightarrow y1}$ when the coupling strength $\varepsilon \geq 0.2$, which is

physically not correct (Fig. E). For 60 samples per pseudo-period, both influences are comparable for larger coupling strengths (Fig. F). From 80 samples per pseudo-period, $T_{x1 \to y1}$ > $T_{y1 \to x1}$ (Figs. G-H), which is physically correct, and one needs to reach ~ 6000 samples per pseudo-period to have $T_{y1 \to x1} \approx 0$ (Fig. I). Thus, results depend on the number of samples per pseudo-period as found by the reviewer, but more importantly they depend on the use of the multivariate formula (Liang, 2021) instead of the 2D formula (Liang, 2014).

In summary, we agree with the reviewer that the assumption of linearity is a limitation of the Liang's approach, which we now emphasize in Section 3.5. However, the tests we did with the CMI method (3D case) confirm the validity of our results (these results have been added to Section 3.5), and the tests with the unidirectionally coupled Rössler systems highlight the "strength" of the multivariate approach versus the bivariate approach. Thus, we are confident in our results. We also mention in Section 3.5 and at the end of the Conclusions that further research is needed in comparing the Liang's approach to other causal methods.

[Figure]

*Figure A. Causal influence measured by the conditional mutual information (CMI) (a) from sea-surface temperature (SST) to turbulent heat flux (THF) and (b) from THF to SST, based on J-OFURO3 satellite observations, when using a time lag.*

[Figure]

***Figure B. Causal influence measured by the conditional mutual information (CMI) (a) from sea-surface temperature (SST) to turbulent heat flux (THF) given SST tendency and (b) from THF to SST given SST tendency, based on J-OFURO3 satellite observations.***

[Figure]

*Figure C. Rate of information transfer (bivariate approach) as a function of the coupling strength applied to unidirectionally coupled Rössler systems with 20 samples per pseudo-period.*

[Figure]

*Figure D. Rate of information transfer (bivariate approach) as a function of the coupling strength applied to unidirectionally coupled Rössler systems with 6000 samples per pseudo-period.*

[Figure]

*Figure E. Rate of information transfer (multivariate approach) as a function of the coupling strength applied to unidirectionally coupled Rössler systems with 20 samples per pseudo-period.*

[Figure]

*Figure F. Rate of information transfer (multivariate approach) as a function of the coupling strength applied to unidirectionally coupled Rössler systems with 60 samples per pseudo-period.*

[Figure]

*Figure G. Rate of information transfer (multivariate approach) as a function of the coupling strength applied to unidirectionally coupled Rössler systems with 80 samples per pseudo-period.*

[Figure]

*Figure H. Rate of information transfer (multivariate approach) as a function of the coupling strength applied to unidirectionally coupled Rössler systems with 200 samples per pseudo-period.*

[Figure]

*Figure I. Rate of information transfer (multivariate approach) as a function of the coupling strength applied to unidirectionally coupled Rössler systems with 6000 samples per pseudo-period.*